# The Impact of Surgical Bowel Preparation on the Microbiome in Colon and Rectal Surgery

**DOI:** 10.3390/antibiotics13070580

**Published:** 2024-06-23

**Authors:** Lauren Weaver, Alexander Troester, Cyrus Jahansouz

**Affiliations:** 1Department of Surgery, University of Minnesota, Minneapolis, MN 55455, USA; weave500@umn.edu (L.W.); troes012@umn.edu (A.T.); 2Division of Colon & Rectal Surgery, Department of Surgery, University of Minnesota, 420 Delaware St. SE, MMC 450, Minneapolis, MN 55455, USA

**Keywords:** mechanical bowel preparation, oral antibiotics, microbiome, surgical outcomes, colorectal surgery

## Abstract

Preoperative bowel preparation, through iterations over time, has evolved with the goal of optimizing surgical outcomes after colon and rectal surgery. Although bowel preparation is commonplace in current practice, its precise mechanism of action, particularly its effect on the human gut microbiome, has yet to be fully elucidated. Absent intervention, the gut microbiota is largely stable, yet reacts to dietary influences, tissue injury, and microbiota-specific byproducts of metabolism. The routine use of oral antibiotics and mechanical bowel preparation prior to intestinal surgical procedures may have detrimental effects previously thought to be negligible. Recent evidence highlights the sensitivity of gut microbiota to antibiotics, bowel preparation, and surgery; however, there is a lack of knowledge regarding specific causal pathways that could lead to therapeutic interventions. As our understanding of the complex interactions between the human host and gut microbiota grows, we can explore the role of bowel preparation in specific microbiome alterations to refine perioperative care and improve outcomes. In this review, we outline the current fund of information regarding the impact of surgical bowel preparation and its components on the adult gut microbiome. We also emphasize key questions pertinent to future microbiome research and their implications for patients undergoing colorectal surgery.

## 1. Introduction

Colorectal surgery is associated with the highest rates of surgical site infection (SSI) among elective surgeries [1]. These infectious complications impose a significant financial burden on the healthcare system and have a profound impact on patient morbidity and mortality [1]. Despite rigorous asepsis surgical measures, SSI rates in elective colorectal surgery range from 5.4% to 23.2%, with a weighted mean of 11.4% [1,2]. Since 2016, the World Health Organization has endorsed the use of preoperative bowel preparation with both oral antibiotics (OAs) and mechanical bowel preparation (MBP) to reduce SSIs in elective colorectal surgery [1,3]. Similarly, the American College of Surgeons and the American Society of Colon and Rectal Surgeons (ASCRS) recommend OAs and MBP to lower SSIs, anastomotic leaks, Clostridium difficile infections, and other surgical complications [4,5]. Although infectious complications are currently at a historic low, it is still unclear why certain patients develop SSIs and suffer anastomotic leaks despite uniform empiric antimicrobial treatment for patients [6].

Some answers to these questions may lie within the colon itself. Recent recognition of the crucial role the gut microbiome plays in maintaining human health and survival has created a new research frontier aimed at further understanding and preventing surgical complications [7]. Although often used interchangeably with microbiota, the term “microbiome” refers to the collection of genomes from all gut microorganisms, including microbial structural elements, metabolites, and the overall environmental condition. The microbiome encompasses the gut microbiota, the assemblage of microbes present in a defined environment [8,9]. Perturbations in the gut microbiota, catalyzed by various factors, including dietary changes, parenteral or enteral antibiotic use, and high-volume lavage from mechanical bowel preparation, result in a phenomenon known as dysbiosis [10]. Although bowel preparation has been around for over one hundred years, its precise mechanism of action and its effects on the human gut microbiome remain elusive [6].

Over the past decade, emerging data have shed light on the impact of bowel preparation on the colonic microbiome. Several studies have demonstrated significant differences in stool and mucosal microbial profiles following MBP, including a decrease in *Lactobacillaceae*, a microbial family typically associated with positive gut health [11,12,13]. Surgical bowel preparation (SBP), consisting of OAs and MBP, has been shown to alter microbiome composition more than MBP alone in colonoscopy patients, although the implications of this are unknown [14]. There has been growing interest and an increase in research published on the effects of SBP on the microbiome, yet a comprehensive understanding of its mechanism of action is lacking. Currently, there is no expert consensus on the microbiota that should be preserved or neutralized to prevent infectious complications [6]. These studies are often limited by small sample sizes, and their conflicting conclusions have left researchers and clinicians with more questions than answers [11,12,13,14].

This review aims to describe the current fund of knowledge regarding the impact of bowel preparation on the human gut microbiome in colorectal surgery. Additionally, it will underscore key questions pertinent to future microbiome research and their implications for surgical outcomes after colorectal surgery.

## 2. Bowel Preparation

### 2.1. Evolution of Bowel Preparation

In the early 20th century, bowel preparation, also referred to as “intestinal asepsis”, emerged in response to elevated rates of infectious complications among patients undergoing colorectal surgery [15]. During this period, postoperative mortality rates ranged from 10 to 12%, with infectious complications being the primary cause. Meanwhile, over 20% of patients experienced an anastomotic leak, and of the patients who survived the surgery, 80–90% suffered from a wound infection [16]. To mitigate these alarming infection rates, MBP was introduced, which included starting patients on a special diet and/or laxatives, in hopes that reducing fecal bacterial load would decrease surgical field contamination and minimize subsequent infectious complications [15]. An additional proposed benefit was that eliminating hard feces would prevent undue pressure and ischemia on the new colonic anastomosis and consequently reduce anastomotic leak rates [17]. Despite conflicting evidence for the efficacy of MBP in reducing bacterial counts, its routine adoption became entrenched in colorectal surgery practice by the 1970s [15,18]. This practice persisted into the modern surgical era, with various combinations of dietary restrictions, enemas, and large-volume saline irrigation via a nasogastric tube. The introduction of polyethylene glycol made MBP more tolerable for the patient by reducing bowel preparation time and electrolyte derangements seen with saline irrigation [15,19]. Although strong evidence behind the use of MBP was sparse, a 1996 national survey of 471 North American colorectal surgeons found that 100% of them endorsed the use of MBP [20,21].

The concurrent advancements in OA regimens further substantiated the use of MBP, as patients who participated in clinical trials evaluating different OA regimens also underwent MBP [22]. Cohn et al. emphasized the three best qualities of an OA regimen: strong bactericidal activity against colonic bacteria, prevention of pathogenic microorganism overgrowth, and a satisfactory patient tolerance profile with low GI tract absorption [23]. Since the most commonly isolated bacteria from SSIs after large bowel surgery were *Escherichia coli* and *Bacteroides fragilis*, early clinical trials evaluated OA efficacy by its ability to eliminate all aerobic and anerobic gut microflora [24]. Some of the first studied OAs were sulfonamide derivatives. However, sulfonamides were deemed an “unsatisfactory agent,” since they failed to eliminate the “streptococci, enterococci, coliform organisms, and bacteriodes” in bacteriologic stool studies [23,25]. Sellwood et al. found that 48 h of preoperative treatment with neomycin and bacitracin was superior to phthalylsulphathiazole, showing a significant reduction in streptococci and coliform bacteria. Interestingly, both antibiotics also reduced quantities of *Bacteroides fragilis*, which these agents are not known to be bactericidal against [26]. One review noted that these results emphasize “our lack of knowledge regarding the interaction of bacterial species and their possible dependence on each other for survival in nature” [24]. Researchers are still unraveling the complexities of gut bacterial interactions, and our understanding of the gut microbiome’s dynamics continues to evolve.

In the 1970s, several clinical trials were conducted on various combinations of OAs in surgical bowel preparation, aiming to further reduce colonic bacterial counts and SSIs [25,27,28]. A notable clinical trial by Nichols et al. compared surgical outcomes and differences in colonic flora amongst elective colorectal surgery patients who underwent MBP followed by preoperative erythromycin–neomycin or no OAs [27]. The results were impressive. Despite the high prevalence of SSIs after colorectal surgery during this era, none of the 20 randomized patients contracted a wound infection, nor did any of the 69 retrospectively analyzed patients who underwent the same preoperative regimen [27]. Furthermore, stool studies revealed the suppression of both anerobic and aerobic bacteria in the erythromycin–neomycin cohort, which the authors attributed as a likely explanation for their lack of SSIs [27]. In addition, Clarke et al. conducted a randomized, double-blind clinical trial on 116 elective colorectal surgery patients who underwent MBP followed by either preoperative neomycin–erythromycin or placebo. The septic complication rate was 9% in the neomycin–erythromycin group compared to 43% in the placebo group [28]. In addition to several other studies, the consensus was that elective colon resection should include both preoperative OA and MBP [27,28].

### 2.2. Mechanical Bowel Preparation Controversy

New evidence emerged regarding the relatively low infection rates following emergency colon surgery, in which patients do not undergo preoperative MBP [29]. This spurred numerous clinical trials in the 1990s and early 2000s, the results of which challenged the prevailing “surgical dogma” regarding the efficacy of MBP in improving surgical outcomes [29,30]. When comparing a mechanically prepped vs. unprepped colon, several studies failed to demonstrate significant differences in surgical outcomes, leading to their conclusion that MBP is not necessary [21,30,31,32]. In one randomized trial, Santos et al. suggested that MBP may even be detrimental, as they observed higher complication rates in the MBP cohort (MBP 21% vs. no MBP 11%, *p* < 0.05), particularly an increase in wound infections (MBP 17% vs. No MBP 9%, *p* < 0.05) [33]. Interestingly, they also analyzed the bacterial composition of the bowel content and peritoneal fluid and found no significant differences in bacterial quantity or composition between patients who underwent MBP and those who did not, which challenged the pre-existing notion that MBP reduces colonic bacterial counts [33]. In a meta-analysis encompassing seven randomized clinical trials conducted between 1992 and 2005, Bucher at al. found that elective colorectal surgery patients who received MBP were more likely to suffer an anastomotic leak (MBP 36/642 vs. no MBP 18/655; OR 1.84; *p* = 0.03) [34]. Infection rates were also higher in the MBP group but failed to reach statistical significance [34]. Lastly, a Cochrane review by Guenaga et al. pooled 5805 clinical trial participants who did or did not receive MBP prior to elective colorectal surgery. Based on their measured outcomes of anastomotic leak and wound infection, they concluded that there was no significant evidence to support the benefit of MBP for patients undergoing colorectal surgery, and it can be “safely omitted” [35].

The controversial evidence related to MBP has caused contention amongst international guidelines pertaining to the optimal bowel preparation prior to elective colorectal surgery. The Enhanced Recovery After Surgery (ERAS) Society and the United Kingdom’s National Institute for Health and Care Excellence recommend against the routine use of MBP [36,37]. Meanwhile, the 2018 WHO, 2019 ASCRS, and 2021 Japan Society for Surgical Infection guidelines support the combined use of OAs along with MBP prior to elective colorectal surgery [3,5,38].

A major criticism of past clinical trials assessing SBP is the lack of granular data between colon versus rectal surgery. Rectal surgery patients, who have often received neoadjuvant chemoradiation, incur a higher SSI rate, particularly in the setting of low colorectal or coloanal anastomosis [39]. Past clinical trials also did not account for the use of minimally invasive surgical (MIS) techniques, which have further decreased SSIs and are now the gold standard for many elective colon and rectal surgeries [40]. One retrospective study using the National Surgical Quality Improvement Program (NSQIP) database compared SSI rates between MIS colon and rectal surgeries [41]. Amongst 12,417 rectal cancer patients, superficial and organ space infection rates were similar; however, the combined OA and MBP group had a significantly lower rate of deep surgical infections (0.1% vs. 0.9% *p* = 0.004). For colon cancer resection, the OA and MBP group incurred lower rates of superficial SSIs (1.1% vs. 1.9% *p* = 0.043), but deep and organ space infections had similar rates [41]. The recently published multicenter, double-blind MOBILE-2 trial randomized 565 patients who underwent MIS or open rectal surgery to receive MBP followed by either oral neomycin and metronidazole or placebo. The OA and MBP cohort experienced a lower rate of postoperative complications, fewer SSIs, and less anastomotic dehiscence [42]. Although compelling, more research is needed to determine the efficacy of SBP for rectal surgery.

Preoperative bowel preparation, in its various iterations, has been around for almost a century. Yet, the evolving and conflicting evidence surrounding SBP showcases our limited understanding of its true mechanism, the impact on the gut microbiome, and potential association with surgical outcomes. If researchers and clinicians aim to further reduce SSIs, more investigation into the gut microbiome may be a promising area to explore.

## 3. Microbiome

The individual configuration and function of the adult gut microbiota are governed by diet and host factors that regulate and direct microbial growth. Disruptions of this homeostasis, known as dysbiosis, have been increasingly linked to a number of human illnesses [43]. The National Institute of Health’s Human Microbiome Project, which aims to further characterize and comprehend how the microbiome affects human health and disease, has established three important factual properties about the human microbiome: (1) the microbiome is exceedingly complex, (2) the spatial distribution of microbes within the body is nonrandom, and (3) microbes associated with the human body can be beneficial, neutral, or pathogenic [44,45]. Understanding the factors that influence microbiota composition, function, and their roles in particular disease processes or pharmacological interventions requires the utilization of metabolomics, proteomics, and metatranscriptomics [8]. These tools have allowed researchers, particularly within colorectal surgical specialties, to analyze the impact of SBP on the gut microbiota.

In the adult human gastrointestinal tract, the total number of commensal enteric bacteria varies greatly from ~10^2,3^ cells/gram in the jejunum and proximal ileum to ~10^7,8^ in the distal ileum, and ~10^11,12^ within the ascending colon [46]. Human studies have identified 2100 species classified into 12 different phyla with 16S rRNA gene sequencing analyses, demonstrating that more than 90% of bacterial species found within the gut belong to 4 phyla: *Proteobacteria*, *Actinobacteria*, *Firmicutes*, and *Bacteroidetes* [46]. Obligately anaerobic primary fermenters, belonging to the classes *Bacteroidia* (phylum *Bacteroidetes*) and *Clostridia* (phylum *Firmicutes*), dominate the high-density microbial community in the large intestine due to the limited availability of oxygen and nitrates [47,48]. In contrast, *Enterobacteriaceae* (phylum *Proteobacteria*) are found in the small intestine, due to shorter transit time and high bile concentrations, but not the colon [49]. Besides nonrandom spatial distribution, gut microbiota also differs by age. Microbiota diversity in children is dominated by *Akkermansia muciniphila, Bacteroides, Clostridium coccoides* spp., *Clostridium botulinum* spp., and *Veillonella* [50]. However, by age 3, an individual’s gut microbiota becomes comparable to that of adults, largely persisting until people become elderly [51]. At that point, due to poorly understood mechanisms, elderly individuals typically exhibit decreased *Bifidobacterium* and increased *Clostridium* and *Proteobacteria* [52]. In addition to generalized compositional changes observed over time, the *Firmicutes* to *Bacteroidetes* ratio has been previously used to characterize dysbiosis in diseased states [53].

The intestinal microbiota serves a variety of functions critical to wound healing, including energy extraction in the form of short-chain fatty acid (SCFA) production and the modulation of pro- and anti-inflammatory immune responses [54]. Butyric acid is associated with the maintenance of gut barrier function and integrity via the regulation of tight junction proteins [55,56]. The depletion of SCFA, including butyric acid, results in gut and systemic inflammation, delayed wound healing, and gut barrier disruption [57,58]. Our group has previously demonstrated that butyric acid levels decreased by 80% after bowel preparation and surgical intervention [14].

Furthermore, biomarkers reflecting gut permeability have been studied, including zonulin and tight junction proteins zonula occludens-1 (ZO-1) and occludin. Zonulin is a primary regulator of intestinal permeability and is implicated in IBD, autoimmune diseases, obesity, and diabetes [59,60,61]. Enteric pathogens are a main trigger for zonulin release, which causes the displacement of ZO-1 and occludin from the tight junction complex [62]. If disrupted, tight junction proteins are released from the barrier and are detectable in blood, reflecting increased gut permeability [60,63,64].

Finally, the effects of antibiotic-induced disruption of the microbiome in association with alterations to the immune system have been studied in Rhesus macaques [65]. Changes in colonic and systemic immunity were seen in the colon with the recruitment of neutrophils, CD4+ T helper 17 (Th17), which produces IL-17, and T helper 22 (Th22) cells, which are major sources of IL-22. IL-17 promotes neutrophil recruitment and synergizes with IL-22 to activate epithelial defense through the expression of anti-bacterial proteins and peptides. These cytokines are crucial in maintaining mucosal immunity against pathogenic bacteria and promoting epithelial regeneration and repair [66,67,68].

Despite all of the characterized changes, absent intervention, the human gut microbiota is largely stable, with one study reporting that >60% of intestinal microbial strains are retained in their host over the course of 5 years [69]. While dynamic alterations of each individual’s microbiota occur in response to a myriad of inputs, such as dietary influences, tissue injury, host and microbiota-specific byproducts of metabolism, we aim to highlight the compositional gut microbiota changes in response to SBP and its components commonly used by colorectal surgery patients.

## 4. Impact of Bowel Preparation on the Microbiome

Despite being commonly employed and highly debated over decades, SBP, the combination of preoperative OA, intravenous antibiotics, and/or MBP, has a poorly understood mechanism. Designed with the goal to broadly decontaminate the intestine, a recent critique of this extensive microbial depletion is that perturbations in the diverse and commensal microbial community can lead to compositional shifts favoring the proliferation of pathologic genera that can negatively impact wound healing and recovery [70].

Previous interventional studies have demonstrated gut microbial sensitivity to antibiotics and bowel surgery [71,72,73,74]. Dethlefsen et al. demonstrated that broad-spectrum antibiotic therapy resulted in long-lasting, although not permanent, alterations in intestinal colonization profiles [72]. In addition, Croswell et al. observed that antibiotic administration led to an increased susceptibility to pathogenic organisms, such as *Salmonella enterica* [75]. Fang et al. compared inflammatory bowel disease (IBD) patients undergoing ileocolonic resection or colectomy to no surgery and observed decreased alpha diversity in the microbiome and metabolome, as well as an elevated relative abundance of *E. coli* in surgery patients [76]. In a systematic literature review, Ferrie et al. reported detailed outcomes from eight studies that examined the effect of surgery on microbiota alterations in patients with IBD, and they observed several compositional changes with beneficial, harmful, and unclear effects of differing magnitudes [77]. A similar variability of outcomes was observed in four studies looking at the impact of colorectal surgery on the microbiota from the same systematic review [77]. While a link between antibiotic administration as well as surgical intervention and their respective changes on the gut microbiota undoubtedly exists, the current state of the literature highlights an abundance of raw data and a paucity of causal pathways.

Despite studies calling into question the utility of MBP on clinical outcomes, few studies have examined the effects of MBP on microbiota compositional shifts with somewhat equivocal results [12,78,79,80]. O’Brien et al. compared stool samples from colonoscopy patients who completed MBP to controls not undergoing a procedure, finding that a small number of subjects had short-term changes but no variance in composition between the two groups, leading to their conclusion that bowel preparation does not have a lasting effect [80]. Despite this controversial negative result, the bulk of evidence supports that MBP can cause widespread and potentially lasting compositional changes [12,81,82,83]. Drago et al. conducted a diet-controlled study of 10 patients undergoing colonoscopy who received 4 L of polyethylene glycol solution and provided three stool samples: 1 week preoperatively, immediately before colonoscopy, and 1 month post-procedure. In these patients, MBP resulted in significant compositional shifts within the microbiota at phylum, class, and family level between pre- and immediately post-bowel lavage. While most of these changes reverted to baseline at the 1-month mark, a persistent reduction in *Lactobacillaceae* and *Enterobacteriaceae* occurred while an abundance of *Rikenellaceae, Eubacteriaceae,* and *Streptococcaceae* was observed (*p* < 0.05) [12]. From a functional standpoint, the impact of these changes is difficult to interpret. *Lactobacillaceae* are known to foster the development of the immune system, provide barrier function, and act to salvage energy as short-chain fatty acids [84,85]. Meanwhile, *Enterobacteriaceae* include many nosocomial pathogens with substantial antibiotic resistance, and *Streptococcaceae* is associated with increases in fecal proteases, which are known to cause inflammation, disrupt mucosal barrier function, and potentially provide a metabolic advantage for certain bacteria [86,87,88].

While individual components of SBP have been shown to cause compositional changes within the gut microbiota, even fewer studies have examined the impact of SBP as a whole. Our group previously published a longitudinal pilot study comparing MBP alone in patients undergoing colonoscopy versus SBP (OA + MBP) in patients undergoing colon surgery [14]. In line with the previous literature, patients in the MBP alone arm had minimal microbiota alterations with rapid return to baseline. However, the SBP group demonstrated larger shifts that persisted for an average of 1 month, a decrease in α diversity, and an increased abundance of *Lactobacillus*, *Enterococcus*, and *Streptococcus*. These results in patients undergoing colon surgery differ from the published data by Drago et al., [12] potentially suggesting that oral antibiotics and surgical stress, the two additional components of the colon surgery arm, as described by Nalluri-Butz et al., play an important role in long-term microbiome compositional alterations (Figure 1). Taken together, although relative abundances of bacteria pre- and post-intervention provide insight into possible downstream functional impacts, the gut microbiota remains a partially understood and exceedingly complex entity.

## 5. Role of Probiotics in Colorectal Surgery

Paralleling the growing fund of knowledge regarding the human gut microbiome and its impact on colorectal patients, the use of probiotics has been an area of research aimed at reducing infectious complications after colorectal surgery [89,90]. Probiotics may help maintain homeostasis within the gut microbiome, which, as discussed, can be disrupted by SBP. Several clinical trials have demonstrated that probiotics aid in maintaining intestinal integrity, reducing bacterial translocation, and promoting a greater number of beneficial versus pathogenic microorganisms [91,92,93]. A systematic review and meta-analysis of 21 clinical trials comparing perioperative probiotics or synbiotics, which are combinations of probiotics with indigestible food ingredients that stimulate bacterial activity to placebo or standard of care in elective colorectal surgery, found significantly fewer pulmonary and urinary tract infections [90]. Interestingly, there was no difference in anastomotic leaks or wound infections [90]. The use of perioperative probiotics shows promise in reducing postoperative complications. However, there is currently a wide variety of probiotic products available, and the effect of specific probiotic strains on the gut microbiome, as well as surgical outcomes, is unknown.

## 6. Clinical Implications and Areas of Future Study

The current body of clinical evidence regarding the effect of bowel preparation on the microbiome in colorectal surgery is expanding. However, conflicting results have hindered the formation of a clear consensus on targeted interventions for patient care [94]. Another challenge in detecting the effect of microbiome changes on patient outcomes arises from the overall low event rates of complications such as anastomotic leaks and surgical site infections in colorectal surgery. At present, interventions directed at beneficial microbiota alteration should not supersede current recommendations for asepsis practices, as the published literature remains largely exploratory. The translation of microbiome data into impactful clinical changes will require the discovery of causal and modifiable pathways that implicate specific microbiota for targeted intervention. With the advancement in technological capabilities, researchers will be better equipped to study the interactions between the human host and microbiome, paving the way for new therapies geared towards microbiome-precision medicine [95].

## 7. Limitations

Studies of the gut microbiome, by their essence, unintentionally incorporate many confounders that impact the quality of study results, including lifestyle or environmental factors, as well as diet. The impact of pre-surgical fasting, preoperative administration of prebiotics or probiotics, and controlling for the use of bowel preparation or delivery of different antibiotics can all significantly impact study results when examining the microbiome. Additionally, the method of sample preservation and analysis has the potential to alter the results. The gold standard for long-term fecal sample preservation is storage in a −80 °C freezer, and standardization of this process could lead to more generalizable results.

As technology improves, detailed characterization of the microbiota and their components will also be enhanced. At present, comparisons between studies examining similar concepts yet utilizing different analytic techniques can generate unique results that can obscure the narrative [96]. This dilemma, ubiquitous in fields at the forefront of scientific discovery, remains an ever-present issue when attempting to link the human microbiome to an ever-increasing number of disease states.

Future research incorporating methodological rigor to control these confounding variables and standardized analytic techniques is essential as we attempt to define causal pathways linking gut microbiota changes with perioperative surgical care.

## 8. Conclusions

Preoperative bowel preparation has successfully reduced the rates of infectious complications in patients undergoing colorectal surgery, yet, until recently, its impact on the gut microbiome has been overlooked. As our collective knowledge of the gut microbiome continues to evolve, we will face many challenges in translating data into impactful clinical interventions. However, harnessing the power of the gut microbiome holds great promise in helping researchers and clinicians achieve their overarching goal of improving patient outcomes in colorectal surgery.

## Figures and Tables

**Figure 1 antibiotics-13-00580-f001:**
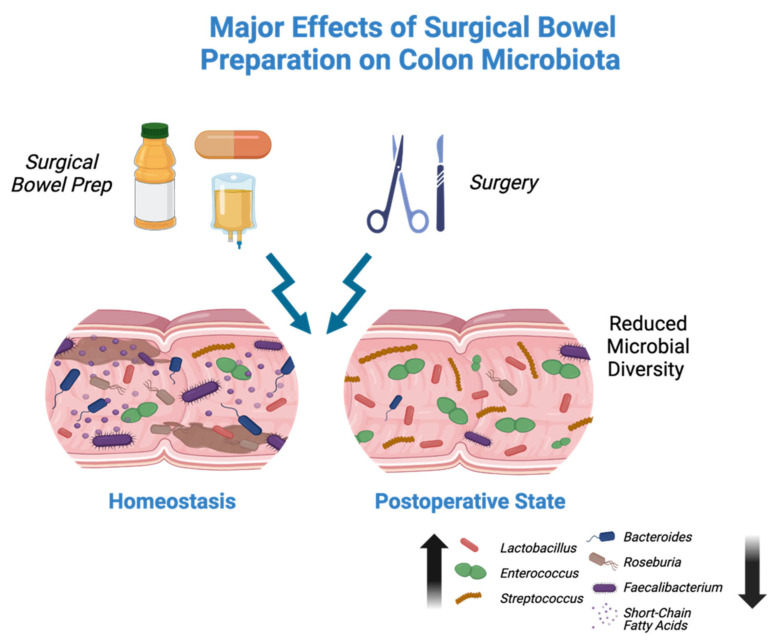
During homeostasis, the colon demonstrates a wide variety of microbial diversity. After preoperative surgical bowel preparation consisting of mechanical bowel preparation, oral antibiotics, and preoperative intravenous antibiotics along with colorectal surgery, postoperative microbial analyses show a reduction in α diversity and microbial-derived metabolites. There is a relative increase in *Lactobacillus*, *Enterococcus*, and *Streptococcus* species, as well as a reduction in *Bacteroides*, *Roseburia*, *Faecalibacterium*, and short-chain fatty acids.

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
