# Peer review of "The Impact of Surgical Bowel Preparation on the Microbiome in Colon and Rectal Surgery"

_antibiotics, 2024, doi:10.3390/antibiotics13070580_

Round 1

Reviewer 1 Report

Comments and Suggestions for Authors

The authors did a good job at writing this review summarizing the research related to surgical bowel preparation and its effect on the gut microbiome and peri-operative care.

This review highlights the importance of considering the outcomes on the microbiome due to surgical bowel preparation. The authors did a good job at compiling studies ranging from how bowel preparation was introduced historically to how the post-surgical recovery may be affected due to the microbiome changes affected by surgical bowel preparation. Further, the authors also included studies related to probiotic administration as a probable solution to this issue.

However, this article could benefit from a pictorial representation of the microbial species affected by surgical bowel preparation.

Author Response

Reviewer #1

The authors did a good job at writing this review summarizing the research related to surgical bowel preparation and its effect on the gut microbiome and peri-operative care.

This review highlights the importance of considering the outcomes on the microbiome due to surgical bowel preparation. The authors did a good job at compiling studies ranging from how bowel preparation was introduced historically to how the post-surgical recovery may be affected due to the microbiome changes affected by surgical bowel preparation. Further, the authors also included studies related to probiotic administration as a probable solution to this issue.

However, this article could benefit from a pictorial representation of the microbial species affected by surgical bowel preparation.

Thank you for the kind words regarding the content of our manuscript. While the abundance of published microbiome research continues at a rapid pace, we have attempted to highlight certain studies pertinent to the components of surgical bowel preparation and their effects on the microbiome. To that end, we have enhanced the figure that accompanied the original submission, where we detail the findings from our own groups previous work on microbial species alterations after surgical bowel preparation.

Reviewer 2 Report

Comments and Suggestions for Authors

The assessed article must be corrected and the information must be expanded. The "Bowel Preparation" chapter is well covered, both historically and controversially.

1. Unfortunately, the "Microbiome" chapter is weak. Most citations concern single bacterial species and not the microbiome. Figure 1 shows only 6 bacteria out of approximately 1,000 found in the intestine. In addition to Bacteroides mentioned, what about the other biggest players in the gut such as Clostridium, Bifidobacterium, Eubacterium, Fusobacterium, Peptostreptococcus, Ruminococcus, Prevotella, Veillonella, and Bacillus?

2. Which metabolites are important, which decrease and which increase?

3. If the authors find works on actual microbiota, preferably examined using the NGS method, these results should be included in the table. 4. Lactobacillus levels in Figure 1 are increasing - is this not related to supplementation? I assume that the patient's physiological Lactobacilli decrease, but because probiotic strains are administered, the Lactobacillus count temporarily increases for a period of 1-2 weeks?

5. Many old publications were cited and their number should be limited. Please add articles about microbiota from the last few years.
